# Short Communication: Differences in Levels of Free Amino Acids and Total Protein in Human Foremilk and Hindmilk

**DOI:** 10.3390/nu10121828

**Published:** 2018-11-26

**Authors:** Joris H. J. van Sadelhoff, Dimitra Mastorakou, Hugo Weenen, Bernd Stahl, Johan Garssen, Anita Hartog

**Affiliations:** 1Danone Nutricia Research, Uppsalalaan 12, 3584 CT Utrecht, The Netherlands; j.h.j.vansadelhoff@uu.nl (J.H.J.v.S.); dimitramastorakou@gmail.com (D.M.); hugo.weenen@danone.com (H.W.); Bernd.stahl@danone.com (B.S.); Johan.garssen@danone.com (J.G.); 2Utrecht Institute for Pharmaceutical Sciences (UIPS), Utrecht University, Universiteitsweg 99, 3584 CG Utrecht, The Netherlands

**Keywords:** breastfeeding, preterm infants, lactation, bioactive factors, glutamine

## Abstract

Free amino acids (FAAs) in human milk are indicated to have specific functional roles in infant development. Studies have shown differences between human milk that is expressed at the beginning of a feed (i.e., foremilk) and the remainder of the milk expressed (i.e., hindmilk). For example, it is well established that human hindmilk is richer in fat and energy than foremilk. Hence, exclusively feeding hindmilk is used to enhance weight gain of preterm, low birthweight infants. Whether FAAs occur differently between foremilk and hindmilk has never been reported, but given their bioactive capacities, this is relevant to consider especially in situations where hindmilk is fed exclusively. Therefore, this study analyzed and compared the FAA and total protein content in human foremilk and hindmilk samples donated by 30 healthy lactating women. The total protein content was found to be significantly higher in hindmilk (*p* < 0.001), whereas foremilk contained a significantly higher total content of FAAs (*p* = 0.015). With regards to individual FAAs, foremilk contained significantly higher levels of phenylalanine (*p* = 0.009), threonine (*p* = 0.003), valine (*p* = 0.018), alanine (*p* = 0.004), glutamine (*p* < 0.001), and serine (*p* = 0.012) than hindmilk. Although statistical significance was reached, effect size analysis of the milk fraction on FAA levels in milk revealed that the observed differences were only small. To what extent these differences are of physiological importance for infant development remains to be examined in future research.

## 1. Introduction

Human milk contains a nutrient composition that supports optimal infant growth and development. Besides nutrients, human milk contains several bioactive proteins that can contribute to the development of the infant’s immune system and play a role in the protective effect of human milk against the development of certain types of allergies, auto-immune diseases and metabolic disorders [1,2,3]. Free amino acids (FAAs), which make up 3–5% of the total amino acids (i.e., protein-bound and free) present in human milk, are increasingly recognized as potential bioactive factors [4]. It was previously shown that levels of specific FAAs, like glutamine and glutamate, increase in human milk during lactation, which might indicate that the occurrence of FAAs in human milk is regulated in time [4,5]. This, in combination with the demonstrated effects of specific FAAs on a variety of tissues and cells, including immune cells, has led researchers to assign functional roles of FAAs in the developing infant. 

Studies have demonstrated that human milk composition is different between milk expressed at the beginning of a feed (foremilk) and that expressed at the end of a feed (hindmilk). It is well established that hindmilk contains higher fat and energy content compared to foremilk [6,7,8]. Hence, exclusively feeding hindmilk is postulated to be the optimal choice for feeding preterm, low birthweight infants [8,9,10]. Apart from differences in fat and energy content, a recent study indicated differences in peptide content between the two milk fractions as well [11]. Specifically, hindmilk contained a higher number of peptides that corresponded to peptides with sequential removal of amino acids from the n- and c-termini than foremilk, possibly due to the fact that hindmilk resides longer in the mammary gland in the presence of native milk exo- and endoproteases than foremilk [11]. This finding suggests a differential release of FAAs from proteins or peptides in foremilk and hindmilk, which potentially leads to differences in the FAA content between the two milk fractions. Based on these findings, we hypothesized that FAAs might be higher in human hindmilk compared to foremilk. Differences in FAA levels between the two milk fractions might alter the availability of bioactive FAAs for the developing infant, which is particularly important to consider in situations where hindmilk is fed exclusively. 

## 2. Methods

### 2.1. Subjects

Thirty healthy lactating mothers, aged 27 to 43 years, participated in the present study. Participating women were on average 23.5 ± 11.3 weeks postpartum (range: 6–44 weeks), were non-smokers, and were either primiparous or multiparous. The average body mass index (BMI) score of the mothers was 23.8 ± 5.2 kg/m² (range: 18.6–38.8 kg/m²). Twenty-seven mothers were of Caucasian ethnicity and three mothers were of another ethnicity. Each of the thirty mothers gave birth to term infants. On average, participants attained a tertiary education level.

### 2.2. Milk Collection 

Participating mothers were instructed to express 30 mL of foremilk and at least 30 mL of hindmilk by pump using only one breast, considering the first 30 mL of milk after milk flow began as foremilk and the remaining milk as hindmilk. The milk samples were immediately warmed to 40 °C in a water bath to ensure constant quality of the samples prior to analyses of total protein content on the same day. Samples of 0.5 mL foremilk and 0.5 mL hindmilk were stored at −20 °C for analysis of the FAA content. 

### 2.3. Milk Sample Analysis

A total of 60 milk samples (i.e., 30 foremilk and 30 hindmilk samples) were analyzed for total protein and FAA content. Total protein content was measured by infrared transmission spectroscopy using the MIRIS Human Milk Analyzer (Mid-IR, MIRIS, Uppsala, Sweden), as described and validated in detail elsewhere [12]. For the analyses of the FAA content, 0.5 mL of each foremilk and hindmilk sample was pretreated with perchloric acid. Levels of FAAs were then measured by an ultra-fast liquid chromatograph system (Shimadzu, ’s-Hertogenbosch, The Netherlands) equipped with an Acquity UPLC BEH C18 column (1.7 m, 100 × 2.1 mm) (Waters, Milford, MA, USA). AA standards (Sigma, Zwijndrecht, The Netherlands) were used for AA peak identification. This method did not permit the detection of proline and cysteine, yielding a total of 18 detectable FAAs as well as taurine.

### 2.4. Statistics

Differences in FAA and protein levels between foremilk and hindmilk samples were analyzed by means of paired sample *T*-tests. Statistical significance was defined as *p* < 0.05 and trends were indicated when *p* < 0.10. All data are reported as means ± SEM. Data were processed and computed for statistics using GraphPad Prism (version 7.03, GraphPad Software Inc., San Diego, CA, USA) and Microsoft Office Excel 2016 (version 16.0.8431.2110, Microsoft Corporation, Redmond, WA, USA).

### 2.5. Ethics

The project proposal, including the consent form used in this study, were submitted to an accredited Medical Research Ethics Committee (the Independent Review Board Nijmegen (IRBN)). The IRBN confirmed that this study did not need a formal ethical committee review according to the Dutch law. This study was conducted according to the guidelines laid down in the Declaration of Helsinki. Written informed consent was obtained from all mothers prior to the study.

## 3. Results

Levels of FAAs and protein in human foremilk and hindmilk samples are shown in Figure 1. Whereas the protein content was significantly higher (*p* < 0.001) in hindmilk compared to foremilk, the total FAA content was significantly higher (*p* = 0.015) in foremilk (Figure 1A). With regards to individual FAAs, foremilk contained significantly higher levels of essential amino acids (EAAs) phenylalanine (*p* = 0.009), threonine (*p* = 0.003), and valine (*p* = 0.018), and of non-essential amino acids (NEAAs) alanine (*p* = 0.004), glutamine (*p* < 0.001), and serine (*p* = 0.012) compared to hindmilk (Figure 1A). Additionally, foremilk had a tendency towards higher levels of EAAs histidine (*p* = 0.092) and leucine (*p* = 0.073) as well as NEAAs asparagine (*p* = 0.080), aspartate (*p* = 0.057), and glutamate (*p* = 0.064) than hindmilk. As represented by the forest plot, the difference in protein content between the two milk fractions can be considered moderate (Hedges’ g = 0.64) whereas the differences in individual FAAs as well as the total FAA content were small (Hedges’ g < 0.5) (Figure 1A). Pearson correlation revealed that levels of FAAs and the total protein content in the milk samples, as well as the effect sizes of the observed differences were independent of the maternal BMI.

Figure 1B,C show the total levels of FAAs and protein in foremilk and hindmilk samples of each subject, plotted against the subjects’ timepoint of lactation at the time of milk sampling. Differences in total FAA and protein content between foremilk and hindmilk samples seemed to be independent on the timepoint of lactation (Figure 1B,C). However, within-subject differences between foremilk and hindmilk appear to be smaller than between-subject differences, for both the total level of FAAs (Figure 1B) and protein (Figure 1C). At the time of milk sampling, the different subjects were at varying stages of lactation. To prevent this from being a driver for between-subject differences, Figure 1B,C only show data of subjects who were at 12 weeks postpartum or later at the time of milk sampling, as FAA and protein levels in human milk are known to remain stable after 3 months of lactation. 

## 4. Discussion

It is discussed that FAAs in human milk have functional roles in infant development besides serving as building blocks for protein synthesis. Studies revealed that human hindmilk is richer in fat and energy than foremilk [6,7,8], and recommendations are made to exclusively feed hindmilk to preterm low birthweight infants to enhance weight gain [8,9,10]. However, whether the FAA content differs between milk fractions has never been reported but might be relevant given their demonstrated bioactive capacities. This study analyzed the FAA and protein content in human hindmilk and foremilk samples of 30 healthy women and reported significant differences in both the total content of FAAs as well as several individual FAAs between the two milk fractions. 

The present study demonstrated significantly higher protein levels in hindmilk than in foremilk. Almost all study participants had higher protein levels in their hindmilk than in foremilk. This finding is both consistent [13,14,15] and inconsistent [16,17,18] with previous studies. A possible explanation for this inconsistency is the inter-study variation in what is considered foremilk and what is considered hindmilk. For instance, some define foremilk as the first 30 mL of milk collected and the remainder as hindmilk, whereas others define foremilk as the milk collected during the first 3 min of the milk flow and the remainder as hindmilk [13,14,15,16,17,18]. With regards to FAAs, this study revealed that hindmilk contained a lower total content of FAAs than foremilk. This is surprising, as hindmilk resides longer in the mammary gland in the presence of active endo- and exopeptidases that cleave of amino acids from milk peptides, a process which may lead to an increase in FAA levels in hindmilk [19,20]. Of note, human milk also contains protease inhibitors which might inhibit the endo- and exopeptidases, however, the interplay between protease inhibitors and proteases in human milk is not well understood [21]. The FAAs that were most responsible for the observed differences were alanine, glutamine, phenylalanine, serine, threonine, and valine, which are all uncharged AAs. However, given the tendency of most of the other FAAs to also be more abundant in foremilk, the difference is unlikely to be amino acid-specific. 

The nutrient composition of human milk is considered insufficient for preterm infants with a low birthweight [22,23]. As human hindmilk is more energy dense than foremilk, exclusively feeding hindmilk to preterm infants is used in practice to obtain a higher rate of weight gain [23,24]. The present study might indicate that this practice provides the preterm infant with higher levels of protein, which might suite the increased protein requirements of preterm infants to achieve improved growth and a normal neurodevelopment [25]. On the other hand, feeding hindmilk exclusively provides preterm infants with lower levels of FAAs. An adequate supply of FAAs is critical for an optimal infant development [26,27,28] and a consistent body of evidence demonstrates that, besides protein, preterm infants also need increased levels of FAAs for optimal growth and neurodevelopmental outcomes [25]. Furthermore, adequate levels of specific FAAs, most notably but not limited to glutamine and glutamate, are proposed to be critical for an optimal development of the immune system and the intestines. Multiple mechanisms are proposed by which glutamine and glutamate contribute to the development of the neonatal immune system. These mechanisms include the modification of T cell and B cell proliferation, the suppression of pro-inflammatory cytokine production by a variety of immune cells, the stimulation of regulatory T cell development and the provision of energy to immune cells [29,30,31,32,33]. It is not surprising, therefore, that a deficiency of glutamine leads to immune impairments, including a reduction in lymphocyte proliferation, an impairment of the expression of surface activation proteins and cytokines, and an induction of apoptosis in a variety of immune cells [29,33]. Similarly, multiple mechanisms have been proposed by which glutamine and glutamate contribute to intestinal development. For instance, they are important energy substrates for intestinal cells and they are involved in the formation of proteins (e.g., tight-junction proteins and glutathione) that are critical for the maturation of the neonatal gut barrier [34,35,36,37]. An insufficient supply of glutamine or glutamate might lead to increased gut permeability and decreased functioning of the mucosa and its associated immune functions [38,39,40]. Preterm infants with a low birthweight, which are under severe stress and generally have a higher intestinal permeability than term infants [41,42], might need higher levels of glutamine and glutamate for an optimal intestinal barrier function [38,43,44]. However, although this study revealed lower FAA levels in hindmilk than in foremilk, the differences were only small. Moreover, inter-individual differences in FAA levels in the milk fractions were profoundly larger than the intra-individual differences between the milk fractions. Whether these minor differences are of physiological importance for infant development needs to be studied in future research. In addition, the findings of this study warrant the investigation of milk of mothers who delivered premature infants and aim to feed their own milk instead of donor human milk, which is usually milk of mothers of term infants.

## 5. Conclusions

The present study demonstrated that the total protein content was significantly higher in hindmilk, whereas the total FAA content was significantly higher in foremilk. Most of the FAAs were more abundant in foremilk, but statistical significance was only reached for phenylalanine, threonine, valine, alanine, serine, and glutamine, some of which have been discussed to play important roles in infant development. Although significant, the observed differences were only small and between-subject differences appear to be larger than within-subject differences. Whether these minor differences have physiological relevance in infant development should be studied further. In addition, the findings of this study warrant the investigation of milk of mothers who delivered premature infants.

## Figures and Tables

**Figure 1 nutrients-10-01828-f001:**
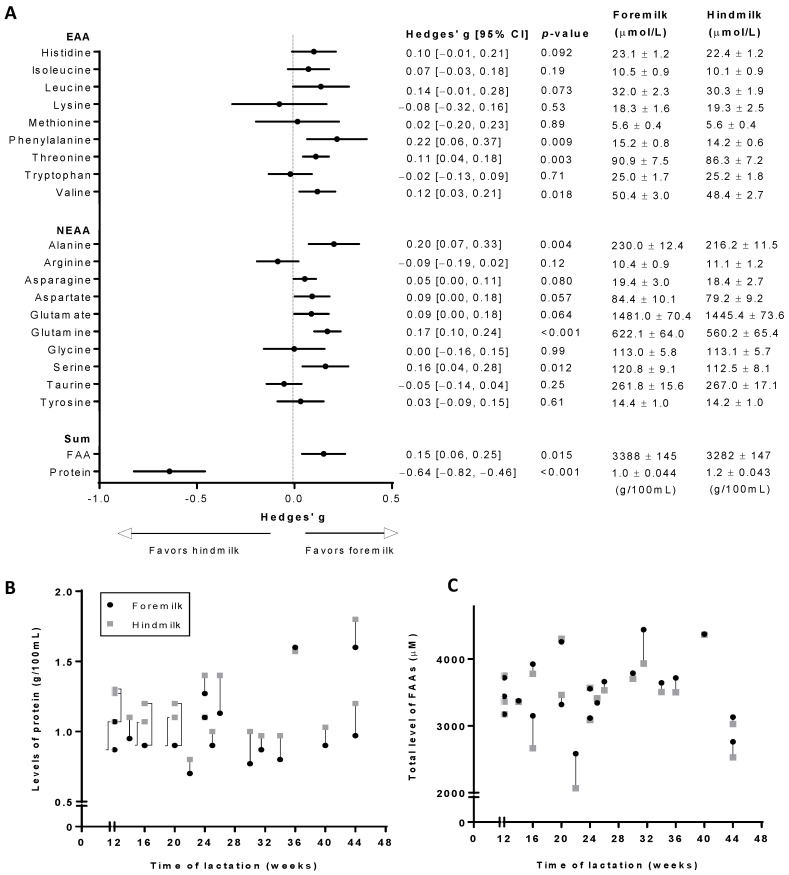
FAA levels in foremilk and hindmilk of thirty women. (**A**) Left: Forest plot indicating the effect sizes (Hedges’ g) of the milk fraction on the FAA and protein content in human milk. Right: average levels of FAAs and the protein content in foremilk and hindmilk samples. For each subject, total levels in hindmilk and foremilk are shown for FAAs (**B**) and protein (**C**), according to the stage of lactation of the subjects at the time of milk sampling. Average levels of FAAs in foremilk and hindmilk are shown as mean ± SEM. FAA: free amino acid; EAA: essential amino acid; NEAA: non-essential amino acid.

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
