# Peer review of "Short Communication: Differences in Levels of Free Amino Acids and Total Protein in Human Foremilk and Hindmilk"

_nutrients, 2018, doi:10.3390/nu10121828_

Round 1
Reviewer 1 Report
There are three issues:
Maternal BMI was not addressed, hence your observations may be different for lean vs obese mothers.
The postpartum range for collection is to great 6-44 weeks (your talk 1 1/2 months - almost 1 y ear) in my view this is a major concern.
A bigger issue for me frankly is this whole idea of foremilk vs hindmilk --- you defined it as the first 30 ml and then the second 30 ml. In my view this is fundamentally wrong. I mean, lets say a mom only has 30 ml of milk at one time, given your definition you are saying she only gave to her infant foremilk -- vs. using volumetric properties or denstitometric principles to define fore-vs-hind milk.
Author Response
Dear reviewer, Thank you for your comments. Please find below my response to each of your comments.
1. Maternal BMI was not addressed; hence your observations may be different for lean vs obese mothers.
Response:
Thank you for observing the lack of information regarding maternal BMI in the manuscript. We documented the BMI values for each of the participants at the time of milk sampling, these have now been added in the material and methods section of the revised manuscript (line 63-64).
Regarding your comment stating that our observations might be different for lean versus obese mothers, we tested whether BMI was associated with protein and free amino acid (FAA) levels in either foremilk, hindmilk or foremilk and hindmilk combined. No associations were found. This statement has been added to the manuscript in the results section (Line 108-110). Also, as best observed for the total protein content represented in Figure 1B, each of the mothers had a slightly higher protein content in hindmilk than in foremilk, indicating that the direction of the effect is not dependent on the maternal BMI. Of critical note, the sample size of the present study might be too low to find any existing correlations between maternal BMI and FAA and protein levels in the milk samples. However, other studies using larger sample sizes did not find any correlations between maternal BMI and FAA or protein levels in human milk either (for references, please see: (1) Kugananthan S et al. Associations between Maternal Body Composition and Appetite Hormones and Macronutrients in Human Milk. Nutrients. 2017 Mar; 9(3): 252. (2) Panagos PG et al. Breastmilk from obese mothers has pro-inflammatory properties and decreased neuroprotective factors. J Perinatol. 2016 Apr (4):284-90. (3) Klein K et al. Concentration of free amino acids in human milk of women with gestational diabetes mellitus and healthy women. Breastfeed Med. 2013 Feb;8(1):111-5.)
2. The postpartum range for collection is to great 6-44 weeks (your talk 1 1/2 months - almost 1 year) in my view this is a major concern. Response:
It is well established in the literature that levels of protein and FAAs in human milk remain stable after 3 months (12 weeks) of lactation. Therefore, whether the samples are collected at 12 weeks or at 44 weeks of lactation should not influence the protein or FAA levels in the milk samples. Only 3 out of the 30 participants collected milk samples prior to 12 weeks of lactation. Excluding any values of these mothers did not alter the statistical outcomes of the study.
In our view, the broad postpartum range for sample collection of the present study is beneficial, as it provided the opportunity to discuss whether the differences observed were dependent on the maternal time of lactation. As depicted in Figure 1B and Figure 1C, the differences in FAA levels and total protein content between foremilk and hindmilk samples did not seem to be dependent on the time of lactation. This is supported by the absence of a correlation of the observed differences (both in absolute and in relative sense) with the time of lactation at sample collection, as analyzed by Pearson correlation. This further reinforces that the broad range for sample collection is not a driver for any of the observed effects.
3. A bigger issue for me frankly is this whole idea of foremilk vs hindmilk --- you defined it as the first 30 ml and then the second 30 ml. In my view this is fundamentally wrong. I mean, let’s say a mom only has 30 ml of milk at one time, given your definition you are saying she only gave to her infant foremilk -- vs. using volumetric properties or densitometric principles to define fore-vs-hind milk.
Response:
To date, there is unfortunately no golden standard for isolating foremilk and hindmilk. We defined foremilk as being the first 30mL of milk after the milk flow began and the remainder as hindmilk. Thus, hindmilk is not only the second 30mL of milk but the milk starting from 30mL, all the way to the ending of the milk flow. This definition is based on a study performed by McDaniel et al. (for reference, please see: McDaniel MR, Barker E, Lederer CL. Sensory characterization of human milk. J Dairy Sci. 1989 May;72(5):1149-58). Some studies use similar definitions as in our study, whereas others use definitions based on the time after the feeding began. As milk flow rates differ largely between lactating mothers (range: 0,86 – 37 mL/min), differentiation between the two milk fractions based on the time after the milk flow began comes with its limitations as well. In the discussion section of our manuscript (line 142-146 in the revised manuscript) we addressed the issue of inconsistency in the differentiation between foremilk and hindmilk among studies as a potential cause of the inconsistency in reported protein levels in the two milk fractions.
In the present study, each of the participating mothers was able to express at least 60ml of breastmilk (the first 30mL being considered as foremilk), excluding the possibility that the first 30mL of milk was the total milk yield. Moreover, let’s say another study considers the first 40 mL of milk as foremilk and the remainder as hindmilk. This would mean that, based on their definition, our hindmilk sample (starting after 30 mL of milk flow onwards) is a mix of foremilk and hindmilk. This would dilute/lower the differences observed when comparing the two different milk samples. Therefore, should the milk sample that is considered hindmilk using our definitions actually be a mixture of foremilk and hindmilk, the differences reported in our manuscript would be underestimated rather than overestimated. Thus, it is highly unlikely that the reported differences between the milk fractions are driven by definitions we used to differentiate between foremilk and hindmilk.
All modifications done in the manuscript are done with ‘track changes’.
I hope my responses to your suggestions are clear.
Sincerely yours,
Anita Hartog
Reviewer 2 Report
This ms very elegantly shows that human foremilk and hindmilk (i.e. the first 30ml of a feeding interval and 30ml from the last minutes) differ in peptide composition and the amounts of FAA (free amino acids). Most interestingly, the authors demonstrate that within-subject differences between foremilk and hindmilk were found to be smaller than between- subject differences for both total FAAs and protein. To even amend the very sound publication I’ve got these suggestions to make.
- Please provide a short definition of foremilk and hindmilk in the abstract to ease it for readers who are not too familiar with the field
- Line 51: this is a very interesting point. If possible, please elaborate on this by referring to the proposed mechanism.
- Line 128: recommendations…are made
- Line 134ff: very interesting! Is there something known like a selective cleavage of amino acids?
- Line 160: Pls. elaborate a little more (which amino acids, what kind of experimental data are out there etc.)
- Please elaborate more why solely feeding hindmilk to premature infants is, according to your new data, not a beneficial way to feed them
Author Response
Dear reviewer, Thank you for your kind comments and suggestions. Please find below my response to each of your suggestions.
1. Please provide a short definition of foremilk and hindmilk in the abstract to ease it for readers who are not too familiar with the field
Response: The following sentence has been added to the abstract (line 15-16) in response to your suggestion: studies have shown differences between milk that is expressed at the beginning of a feed (i.e. foremilk) and the remainder of the milk (i.e. hindmilk).
2. Line 51: this is a very interesting point. If possible, please elaborate on this by referring to the proposed mechanism.
Response: The statement in line 51 was not based on literature showing that that the cleavage of amino acids from peptides contributes to the FAA levels in human milk but was rather a speculation deducted from the previous sentences. Namely, studies have shown that hindmilk contains a higher count of peptides with sequential removal of amino acids from the N- and C-termini than foremilk. The removal of amino acids from human milk peptides by proteases releases free amino acids (FAAs), which would most likely end up in the milk. Therefore, we hypothesized that FAAs might be differentially present in foremilk and foremilk. This concept has, to the best of our knowledge, not been reported earlier. Hence, mechanistic studies remain to be performed.
For clarification purposes, lines 53-54 (formerly line 51) in the revised manuscript have been slightly altered.
3. Line 128: recommendations…are made
Response: Thank you for providing this correction. The word ‘recommendation’, now in line 134, has been changed to ‘recommendations’.
4. Line 134: very interesting! Is there something known like a selective cleavage of amino acids? Response: Human milk contains a variety of exo- and endopeptidases, which each have different preferences regarding the cleavage of specific amino acids. Therefore, we could speculate which amino acids are more likely to be cleaved off than others. For instance, one of the known exopeptidases that might be responsible for the selective cleavage of FAAs from milk proteins is carboxypeptidase B2, which cleaves of basic amino acids arginine and lysine. Based on the fact that hindmilk resides longer in the mammary gland in the presence of this peptidase, we could hypothesize that hindmilk contains higher levels of free arginine and lysine than foremilk. Although arginine and lysine were the only two FAAs that seemed to be slightly more abundant in hindmilk than in foremilk, these differences were not significant. Of note, human milk also contains protease inhibitors which selectively inhibit certain proteases. This process will also influence the rate by which specific amino acids are cleaved off. However, the interplay between proteases and their inhibitors in human milk is not fully understood yet.
Also, specific proteins in human milk seem to be more susceptible to proteases present in human milk than others. This differential susceptibility is proposed to be dependent on the structure of the protein: the more complex the protein structure, the less susceptible the protein is for protease activity.
5. Line 160: Pls. elaborate a little more (which amino acids, what kind of experimental data are out there etc.)
Response: Thank you for the request to elaborate on this section of the manuscript. Lines 167-178 in the revised manuscript provide the elaboration as requested.
6. Please elaborate more why solely feeding hindmilk to premature infants is, according to your new data, not a beneficial way to feed them
Response: The data in our manuscript show that there are slight but significant differences in total protein content and FAA levels between foremilk and hindmilk samples. Thus, our data indicates that the practice of exclusively feeding hindmilk to preterm infants might provide the infant with slightly lower levels of FAAs. However, the data also shows that inter-subject variation is much larger than the intra-subject differences between foremilk and hindmilk. Therefore, physiological relevance of the observed differences might be limited. Future studies should examine whether these minor differences indeed are only of limited physiological relevance. In summary, our data does not directly indicate a major concern of feeding hindmilk exclusively, but rather warrant additional research into the physiological effects of this feeding practice for the preterm infant.
All modifications done in the manuscript are done with ‘track changes’.
I hope my responses to your suggestions are clear.
Sincerely yours
Round 2
Reviewer 1 Report
Thank you for addressing issues I brought up.